# DATA AUGMENTATION INSTEAD OF EXPLICIT REGULARIZATION

## ABSTRACT

Modern deep artificial neural networks have achieved impressive results through models with very large capacity—compared to the number of training examples—that control overfitting with the help of different forms of regularization. Regularization can be implicit, as is the case of stochastic gradient descent or parameter sharing in convolutional layers, or explicit. Most common explicit regularization techniques, such as dropout and weight decay, reduce the effective capacity of the model and typically require the use of deeper and wider architectures to compensate for the reduced capacity. Although these techniques have been proven successful in terms of results, they seem to *waste* capacity. In contrast, data augmentation techniques reduce the generalization error by increasing the number of training examples and without reducing the effective capacity. In this paper we systematically analyze the effect of data augmentation on some popular architectures and conclude that data augmentation alone—without any other explicit regularization techniques—can achieve the same performance or higher as regularized models, especially when training with fewer examples.

## 1 INTRODUCTION

Regularization plays a central role in machine learning. Loosely defined, regularization is any modification applied to a learning algorithm that helps prevent overfitting and improve generalization. Whereas in simple machine learning algorithms the sources of regularization can be easily identified as explicit terms in the objective function, in modern deep neural networks the sources of regularization are multiple and some of them are not explicit, but implicit.

Although the terms explicit and implicit regularization have been used recently in the literature (Neyshabur et al., 2014; Zhang et al., 2017), their distinction is rather subjective. We propose the following definitions:

- **Explicit regularization** techniques are those *specifically* and solely designed to constrain the effective capacity of a given model in order to reduce overfitting. Furthermore, explicit regularizers are not a structural or essential part of the network architecture, the data or the learning algorithm and can typically be added or removed easily.

- **Implicit regularization** is the reduction of the generalization error or overfitting provided by characteristics of the network architecture, the training data or the learning algorithm, which are not specifically designed to constrain the effective capacity of the given model.

Examples of explicit regularizers are weight decay (Hanson & Pratt, 1989), which penalizes large parameters; dropout (Srivastava et al., 2014), which randomly removes a fraction of the neural connections during training; or stochastic depth (Huang et al., 2016), which drops whole layers instead. Implicit regularization effects are provided by the popular stochastic gradient descent (SGD) algorithm, which tends to converge to solutions with small norm (Zhang et al., 2017); convolutional layers, which impose parameter sharing based on prior knowledge about the data; batch normalization (Ioffe & Szegedy, 2015), whose main goal is reducing the the internal covariate shift, but also implicitly regularizes the model due to the noise in the batch estimates for mean and variance.

Driven by the efficient use and development of GPUs, much research efforts have been devoted to finding ways of training deeper and wider networks of larger capacity (Simonyan & Zisserman,

2014; He et al., 2016; Zagoruyko & Komodakis, 2016), Ironically, their effective capacity is eventually reduced in practice by the use of weight decay and dropout, among other explicit regularizers. It is known, for instance, that the gain in generalization provided by dropout comes at the cost of using larger models and training for longer (Goodfellow et al., 2016). Hence, it seems that with such an approach deep networks are wasting capacity (Dauphin & Bengio, 2013). As a matter of fact, unlike traditional machine learning models, deep neural networks seem not to need explicit regularizers to generalize well, as recently suggested by Zhang et al. (2017).

One popular technique that also improves generalization is data augmentation. Importantly, it differs from explicit regularizers mainly in that it does not reduce the effective capacity of the model. Data augmentation is a very old practice in machine learning (Simard et al., 1992) and it has been identified as a critical component of many models (Ciresan et al., 2010; Krizhevsky et al., 2012; LeCun et al., 2015). However, although some authors have reported the impact of data augmentation on the performance of their models and, in some cases, a comparison of different amount of augmentation (Graham, 2014) the literature lacks, to our knowledge, a systematic analysis of the impact of data augmentation on deep neural networks compared to the most popular regularization techniques.

## 1.1 OUR CONTRIBUTIONS

In this paper, we systematically analyze the role of data augmentation in deep neural networks for object recognition, compare it to some popular explicit regularization techniques, discuss its relationship with model capacity and test its potential to enhance learning from less training data and adapt to different architectures.

### 1.1.1 DATA AUGMENTATION AND EXPLICIT REGULARIZATION

Zhang et al. (2017) recently raised the thought-provoking idea that *explicit regularization may improve generalization performance, but is neither necessary nor by itself sufficient for controlling generalization error*. The authors came to this conclusion from the observation that turning off the explicit regularizers of a model does not prevent the model from generalizing—although the performance does become degraded. This contrasts with traditional machine learning involving convex optimization, where regularization is necessary to avoid overfitting and generalize.

However, Zhang et al. (2017) consider data augmentation an explicit form of regularization comparable to weight decay and dropout. We argue instead that data augmentation deserves a different classification due to some fundamental properties: Notably, data augmentation does not reduce the effective capacity of the model. Explicit regularizers are often used to counteract overfitting, but as a side effect the architecture needs to be larger and the training longer (Krizhevsky et al., 2012; Goodfellow et al., 2016). In contrast, data augmentation increases the number of training examples—although not in an independently distributed way—and the robustness against input variability. This has the welcome side-effect of implicitly regularizing the model and improving generalization.

Here, we build upon some of the ideas and procedures from Zhang et al. (2017) and perform some experiments to assess the role of data augmentation in deep neural networks and in particular in contrast to explicit regularizers (weight decay and dropout). In our experiments, we consider two levels of augmentation, light and heavier, as well as no augmentation at all. Then, we test them on two popular successful network architectures: the relatively shallow all convolutional network net (Springenberg et al., 2014) and the deeper wide residual network (Zagoruyko & Komodakis, 2016), trained on CIFAR-10 and CIFAR-100 (Krizhevsky & Hinton, 2009), with and and without explicit regularization. Our central conclusion can be summarized as:

> *In a deep convolutional neural network trained with sufficient level of data augmentation, optimized by SGD, explicit regularizers (weight decay and dropout) might not provide any additional generalization improvement.*

### 1.1.2 DATA AUGMENTATION AND TRAINING WITH FEWER EXAMPLES

Augmented data might be regarded as artificial and very similar to the source examples, therefore with limited contribution for making a network learn more useful representations. However, it has proven to be very useful in extreme cases such as one-shot learning, where only one or few training examples are available (Vinyals et al., 2016).

In order to provide a better insight of the usefulness of data augmentation, we train the networks with only 80%, 50 %, 10 % and 1 % of the available training data and test the effect of data augmentation, again in contrast to explicit regularizers. The summary of our findings in this regard can be summarized as:

*When a deep neural network is trained with a subset of the training data, heavier data augmentation achieves a smaller gap with respect to the baseline model, especially if no explicit regularization is used. Thus, data augmentation seems to serve as true data to a great extent.*

### 1.1.3 Data augmentation and adaptability

One of the disadvantages of explicit regularization is that the parameters highly depend on the network architecture, the amount of training data and other factors. Therefore, if the architecture or other factors change, one has to tune the regularization hyperparameters to achieve comparable results. In order to analyze how data augmentation adapts to different architectures, we test several augmentation schemes on shallower and deeper versions of the network, with and without explicit regularization. Our finding is the following:

*Data augmentation easily adapts to different depths without tuning its parameters. If no explicit regularization is used, we observe that a shallower network achieves slightly worse results and a deeper architecture achieves better results.*

### 1.2 Related work

Regularization is a central research topic in machine learning as it is a key component for ensuring good generalization (Girosi et al., 1995; Müller, 2012). In the case of deep learning, where networks tend to have several orders of magnitude more parameters than training examples, statistical learning theory (Vapnik & Chervonenkis, 1971) indicates that regularization becomes even more crucial. Accordingly, a myriad of tools and techniques have been proposed as regularizers: early stopping (Plaut et al., 1986), weight decay (Hanson & Pratt, 1989) and other $L^p$ penalties, dropout (Srivastava et al., 2014) and stochastic depth (Huang et al., 2016), to name a few examples. Besides, other successful techniques have been studied for their regularization effect, despite not being explicitly intended as such. That is the case of unsupervised pre-training (Erhan et al., 2010), multi-task learning (Caruana, 1998), convolutional layers (LeCun et al., 1990), batch normalization (Ioffe & Szegedy, 2015) or adversarial training (Szegedy et al., 2013).

Data augmentation is another almost ubiquitous technique in deep learning, especially for computer vision tasks, which can be regarded as an implicit regularizer because it improves regularization. It was already used in the late 80's and early 90's for handwritten digit recognition (Simard et al., 1992) and it has been identified as a very important element of many modern successful models, like AlexNet (Krizhevsky et al., 2012), All-CNN (Springenberg et al., 2014) or ResNet (He et al., 2016), for instance. In some cases, data augmentation has been applied heavily with successful results (Wu et al., 2015). In domains other than computer vision, data augmentation has also been proven effective, for example in speech recognition (Jaitly & Hinton, 2013), music source separation (Uhlich et al., 2017) or text categorization (Lu et al., 2006).

Bengio et al. (2011) focused on the importance of data augmentation for recognizing handwritten digits (MNIST) through greedy layer-wise unsupervised pre-training (Bengio et al., 2007). The main conclusion of that work was that deeper architectures benefit more from data augmentation than shallow networks. Zhang et al. (2017) included data augmentation in their analysis of the role of regularization in the generalization of deep networks, although it was considered an explicit regularizer similar to weight decay and dropout. A few works have reported the performance of their models when trained with different types of data augmentation levels, as is the case of Graham (2014). Recently, the deep learning community seems to have become more aware of the importance of data augmentation and new techniques, such as cutout (DeVries & Taylor, 2017a) or augmentation in the feature space (DeVries & Taylor, 2017b), have been proposed. Very interestingly, models that automatically learn useful data transformations have also been published recently (Hauberg et al., 2016; Lemley et al., 2017; Ratner et al., 2017).

## 2 EXPERIMENTS AND RESULTS

This section describes the experimental setup for systematically analyzing the role of data augmentation in modern deep neural networks and presents the most relevant and interesting results.

### 2.1 SETUP

All the experiments are performed on the neural networks API Keras (Chollet et al., 2015) on top of TensorFlow (Abadi et al., 2015) and on a single GPU NVIDIA GeForce GTX 1080 Ti.

#### 2.1.1 NETWORK ACRCHITECTURES

We perform our experiments on two popular architectures that have achieved successful results in object recognition tasks: the all convolutional network, All-CNN (Springenberg et al., 2014) and the wide residual network, WRN (Zagoruyko & Komodakis, 2016). We choose these networks not only because of their effectiveness, but also because they have simple architectures, which is convenient for drawing clearer conclusions. All-CNN has a relatively small number of layers and parameters, whereas WRN is rather deep and has many more parameters.

**All convolutional net.** All-CNN consists of only convolutional layers with ReLU activations (Glorot et al., 2011), it is relatively shallow (12 layers) and has about 1.3 M parameters. The architecture can be described as follows:

$$2\times96C3(1)\text{--}96C3(2)\text{--}2\times192C3(1)\text{--}192C3(2)\text{--}192C3(1)\text{--}192C1(1)$$
$$\text{--}N.Cl.C1(1)\text{--}Gl.Avg.\text{--}Softmax$$

where $KCD(S)$ is a $D \times D$ convolutional layer with $K$ channels and stride $S$, followed by batch normalization and a ReLU non-linearity. *N.Cl.* is the number of classes and Gl.Avg. refers to global average pooling. The network is identical to the All-CNN-C architecture in the original paper, except for the introduction of the batch normalization layers. We set the same training parameters as in the original paper in the cases they are reported. Specifically, in all experiments the All-CNN networks are trained using stochastic gradient descent with batch size of 128, during 350 epochs, with fixed momentum 0.9 and learning rate of 0.01 multiplied by 0.1 at epochs 200, 250 and 300. The kernel parameters are initialized according to the Xavier uniform initialization (Glorot & Bengio, 2010).

**Wide Residual Network.** WRN is a modification of ResNet (He et al., 2016) that achieves better performance with fewer layers, but more units per layer. Although in the original paper several combinations of depth and width are tested, here we choose for our experiments the WRN-28-10 version (28 layers and about 36.5 M parameters), which is reported to achieve the best results on CIFAR. It has the following architecture:

$$16C3(1)\text{--}4\times160R\text{--}4\times320R\text{--}4\times640R\text{--}BN\text{--}ReLU\text{--}Avg.(8)\text{--}FC\text{--}Softmax$$

where $KR$ is a residual block with residual function BN–ReLU–$K$C3(1)–BN–ReLU–$K$C3(1). BN is batch normalization, Avg.(8) is spatial average pooling of size 8 and FC is a fully connected layer. The stride of the first convolution within the residual blocks is 1 except in the first block of the series of 4, where it is 2 to subsample the feature maps. As before, we try to replicate the training parameters of the original paper: we use SGD with batch size of 128, during 200 epochs, with fixed Nesterov momentum 0.9 and learning rate of 0.1 multiplied by 0.2 at epochs 60, 120 and 160. The kernel parameters are initialized according to the He normal initialization (He et al., 2015).

#### 2.1.2 DATA

We perform the experiments on the two highly benchmarked data sets CIFAR-10 and CIFAR-100 (Krizhevsky & Hinton, 2009), which are labeled according to 10 and 100 object classes respectively. Both data sets consist of 60,000 32 x 32 color images split into 50,000 for training and 10,000 for testing. In all our experiments, the input images are fed into the network with pixel values normalized to the range $[0, 1]$ and with floating precision of 32 bits. So as to analyze the role of data augmentation, we test the network architectures presented above with two different augmentation schemes as well as with no data augmentation at all:

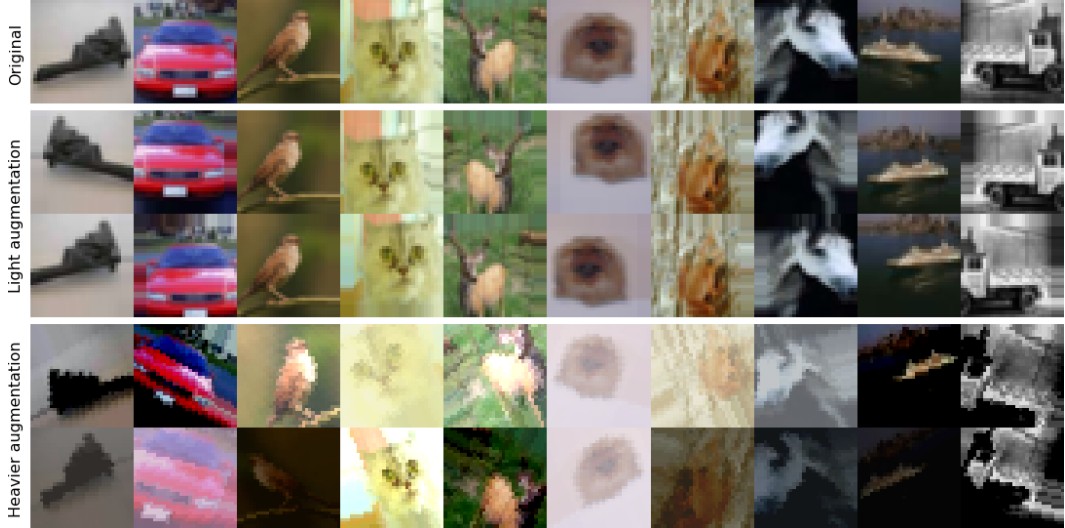

Figure 1: Random images from CIFAR10 transformed according to the augmentation schemes used in our experiments, choosing extreme values from the augmentation parameters. Note that these images are very unlikely to be used during training.

Table 1: Description and range of possible values of the parameters used for the heavier augmentation. $B(p)$ denotes a Bernouilli distribution and $\mathcal{U}(a, b)$ a uniform distribution.

| Parameter | Description | Range |
|-----------|-------------|-------|
| $f_h$ | Horizontal flip | $1 - 2B(0.5)$ |
| $t_x$ | Horizontal translation | $\mathcal{U}(-0.1, 0.1)$ |
| $t_y$ | Vertical translation | $\mathcal{U}(-0.1, 0.1)$ |
| $z_x$ | Horizontal scale | $\mathcal{U}(0.85, 1.15)$ |
| $z_y$ | Vertical scale | $\mathcal{U}(0.85, 1.15)$ |
| $\theta$ | Rotation angle | $\mathcal{U}(-\frac{\pi}{180}22.5, \frac{\pi}{180}22.5)$ |
| $\phi$ | Shear angle | $\mathcal{U}(-0.15, 0.15)$ |
| $\gamma$ | Contrast | $\mathcal{U}(0.5, 1.5)$ |
| $\delta$ | Brightness | $\mathcal{U}(-0.25, 0.25)$ |

***Light* augmentation.** This scheme is adopted from the literature, for example (Goodfellow et al., 2013; Springenberg et al., 2014), and performs only horizontal flips and horizontal and vertical translations of 10% of the image size.

***Heavier* augmentation.** This scheme performs a larger range of affine transformations, as well as contrast and brightness adjustment:

- Affine transformations: $\begin{bmatrix} x' \\ y' \\ 1 \end{bmatrix} = \begin{bmatrix} f_h z_x \cos(\theta) & -z_y \sin(\theta + \phi) & t_x \\ z_x \sin(\theta) & z_y \cos(\theta + \phi) & t_y \\ 0 & 0 & 1 \end{bmatrix} \begin{bmatrix} x \\ y \\ 1 \end{bmatrix}$

- Contrast adjustment: $x' = \gamma(x - \overline{x}) + \overline{x}$

- Brightness adjustment: $x' = x + \delta$

The description and range of values of the parameters are specified in Table 1 and some examples of transformed images with extreme values of the parameters are provided in Figure 1. The choice of the parameters is arbitrary and the only criterion was that the objects are still recognizable, by visually inspecting a few images. We deliberately avoid designing a particularly successful scheme.

## 2.2 A SUBSTITUTE FOR EXPLICIT REGULARIZATION

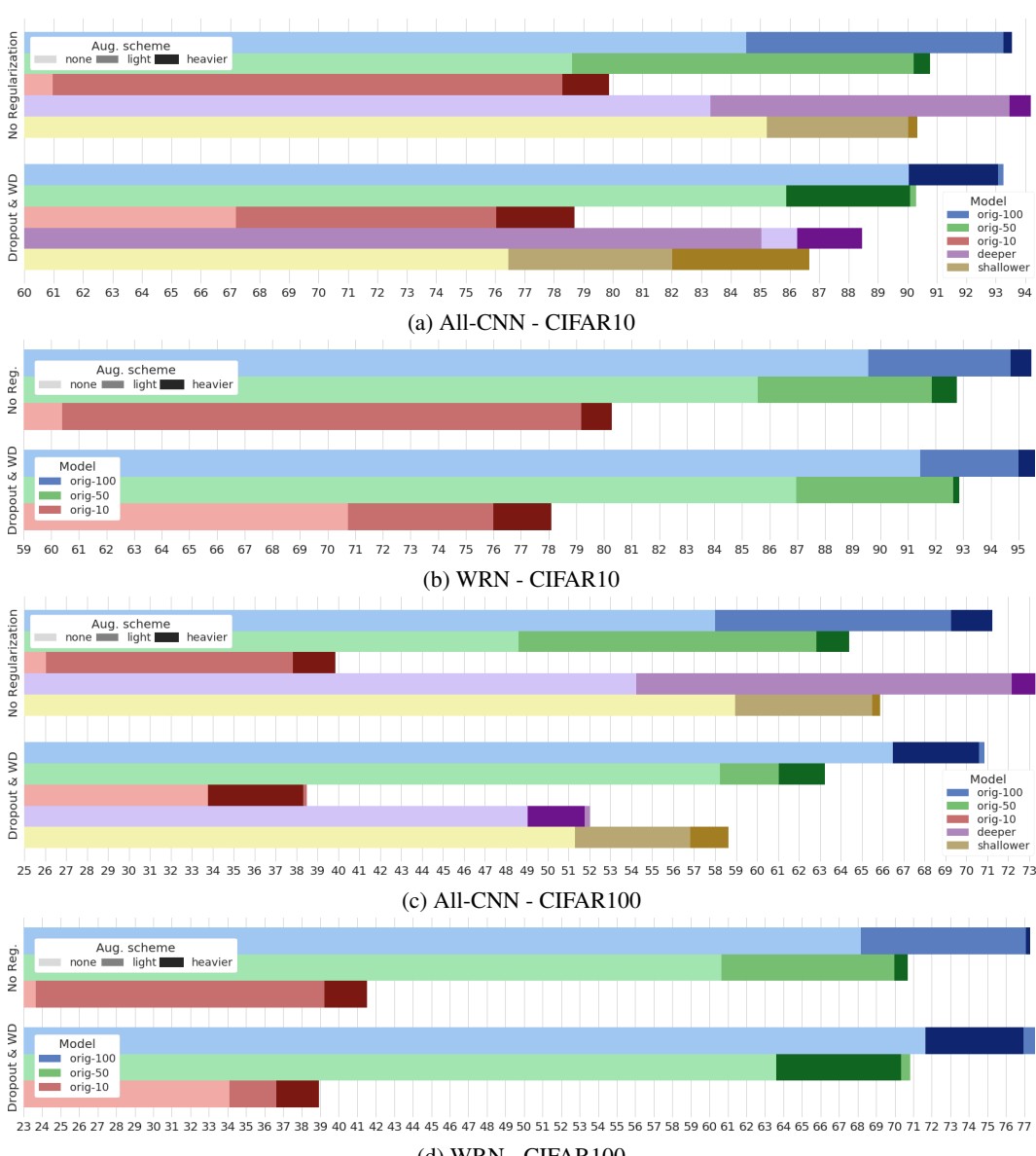

Figure 2: Test accuracy of the networks All-CNN and WRN on CIFAR-10 and CIFAR-100, trained without any explicit regularization (upper groups of bars) and with both dropout and weight decay (lower groups), as in the original papers. The different bars represent different models (original, deeper and shallower) and different percentage of training images (100, 50 and 10 %). The different shades within each bar show the result of training with each data augmentation scheme (none, light and heavier). In most cases, the models trained without regularization achieve the same performance as the explicitly regularized models, or even significantly higher accuracy, as is the case of the shallower and deeper models and when training with fewer examples.

In order to analyze the role of data augmentation and test the hypothesis that it might serve as a substitute for explicit regularization techniques, we first try to replicate the results of All-CNN and WRN provided in the original papers, achieved with both weight decay and dropout. Then, we train the models without weight decay and finally without neither weight decay nor dropout. We test all these different models with the three data augmentation schemes: light, heavier and no augmentation. Additionally, we test the effect of removing the batch normalization (see Appendix A).

As reported by previous works (Krizhevsky et al., 2012; Simonyan & Zisserman, 2014), when the models are trained with data augmentation, at test time slightly better results are obtained by augmenting the test set as well. Therefore, the test accuracy reported here comes from averaging the softmax posteriors over 10 random *light* augmentations.

The main results of the different experiments are shown in Figure 2 (blue bars) and the full report of all experiments can be found in Table 2 of the Appendix A. As expected, both explicit regularization —weight decay and dropout—and data augmentation are successful in reducing the generalization error. However, some relevant observations can be made. Most notably, it seems that data augmentation alone is able to regularize the model as much as in combination with weight decay and dropout and in some cases it clearly achieves better performance, as in the case of All-CNN. Another observation is that without explicit regularization, heavier augmentation always provides better results than light augmentation, whereas with regularization this effect is counterintuitively not consistent.

## 2.3 FEWER AVAILABLE TRAINING EXAMPLES

We extend the analysis of the data augmentation role by training the same networks with fewer training examples. Similarly, we also analyze the combination of data augmentation and explicit regularization in this case. All models are trained with the same random subset of data and tested in the same test set as the previous experiments in order to enable fairer comparisons. Figure 2 (green and red bars) shows the main results with 50 % and 10 % of the available data and the full report, as well as additional experiments with 80 % and 1 % of the data, is given in Table 3 of the Appendix A.

As expected, the performance decays with the number of available training examples. However, as the level of data augmentation increases, the difference with respect to the baseline performance (by training with all examples) significantly decreases. This indicates that data augmentation serves, to a great extent, as true data. Therefore, this confirms the effectiveness of this technique when not many training examples are available.

Furthermore, in all the experiments with a reduced set of the available data, the observations presented above become even clearer. It seems that if explicit regularization is removed, data augmentation alone better resists the lack of data. This can probably be explained by the fact that explicit regularization reduces the effective capacity, preventing the model from taking advantage of the augmented data.

## 2.4 SHALLOWER AND DEEPER ARCHITECTURES

Finally, we perform the same experiments on shallower and deeper versions of All-CNN, so as to analyze how data augmentation and regularization are handled by architectures of different depth. We test a shallower network with 9 layers instead of 12 and 374 K parameters instead of 1.3 M:

$$2\times96C3(1)–96C3(2)–192C3(1)–192C1(1)–N.Cl.C1(1)–Gl.Avg.–Softmax$$

and a deeper network with 15 layers and 2.4 M parameters:

$$2\times96C3(1)–96C3(2)–2\times192C3(1)–192C3(2)–2\times192C3(1)–192C3(2)–192C3(1)–192C1(1)$$
$$–N.Cl.C1(1)–Gl.Avg.–Softmax$$

The results in Figure 2 (purple and brown bars) together with the detailed report of results in Table 4 of the Appendix A show that if the explicit regularization is removed and data augmentation applied, the shallower network achieves slightly worse results and the deeper network slightly better results than the original network. This behavior can be explained by the reduced or increased depth and number of parameters. However, with the explicit regularization active, the results dramatically decrease in both cases. The most probable explanation is that the regularization parameters are not adjusted to the architecture, whereas in the original models the parameters where finely tuned by the authors to obtain state of the art results. This highlights another important advantage of data augmentation: the adjustment of its parameters depends mostly on the training data, rather than on the particular architecture, which offers much more flexibility compared to using explicit regularization. In Appendix B we provide ananalysis of the norm of the weight matrices that helps shed some more light on how the different levels of regularization and data augmentation affect the learned models.

# 3 DISCUSSION AND CONCLUSION

In this work, we have presented a systematic analysis of the role of data augmentation in deep neural networks for object recognition, focusing on the comparison with popular techniques of explicit regularization. We have built upon the work by Zhang et al. (2017), where the authors concluded that explicit regularization is not necessary, although it improves generalization performance. Here, we have shown that it is not only unnecessary, but also that the generalization gain provided by explicit regularization can be achieved by data augmentation alone.

The importance of these results lies in the fact that explicit regularization is the standard tool to enable the generalization of most machine learning methods. However, according to Zhang et al. (2017), explicit regularization plays a different role in deep learning, not explained by statistical learning theory (Vapnik & Chervonenkis, 1971). We argue instead that the theory still holds in deep learning, but one has to properly consider the crucial role of implicit regularization. Explicit regularization is no longer necessary because its contribution is already provided by the many elements that implicitly regularize the models: SGD, convolutional layers or data augmentation, among others.

Whereas explicit regularizers, such as weight decay and dropout, succeed in mitigating overfitting by blindly reducing the effective capacity of a model, implicit regularization operates more effectively at capturing important characteristics of the data (Neyshabur et al., 2014). For instance, convolutional layers successfully reduce the capacity of a model by imposing a parameter sharing strategy that incorporates some essential prior domain knowledge, as well as data augmentation by transforming the training examples in a meaningful and plausible way.

In this regard it is worth highlighting some of the advantages of data augmentation: Not only does it not reduce the effective capacity of the model, but it increases the number of training examples, which, according to statistical learning theories, reduces the generalization error. Furthermore, if the transformations are such that they reflect plausible variations of the real objects, it increases the robustness of the model and it can be regarded as a data-dependent prior, similarly to unsupervised pre-training (Erhan et al., 2010). Besides, unlike explicit regularization techniques, data augmentation does not increase the computational complexity because it can be performed in parallel to the gradient updates on the CPU, making it a computationally free operation. Finally, in Section 2.4 we have shown how data augmentation transparently adapts to architectures of different depth, whereas explicitly regularized models need manual adjustment of the regularization parameters.

Deep neural networks can especially benefit from data augmentation because they do not rely on precomputed features and because the large number of parameters allows them to shatter the augmented training set. Actually, if data augmentation is included for training, we might have to reconsider whether deep learning operates in an *overparameterization* regime, since the model capacity should take into account the amount of training data, which is exponentially increased by augmentation.

Some argue that despite these advantages, data augmentation is a highly limited approach because it depends on some prior expert knowledge and it cannot be applied to all domains. However, we argue instead that expert knowledge should not be disregarded but exploited. A single data augmentation scheme can be designed for a broad family of data, e.g. natural images, and effectively applied to a broad set of tasks, e.g. object recognition, segmentation, localization, etc. Besides, some recent works show that it is possible to learn the data augmentation strategies (Lemley et al., 2017; Ratner et al., 2017) and future research will probably yield even better results in different domains.

Finally, it is important to note that, due to computational limitations, we have performed a systematic analysis only on CIFAR-10 and CIFAR-100, which consist of very small images. These data sets do not allow performing more agressive data augmentation since the low resolution images can easily show distortions that hinder the recognition of the object. However, some previous works (Graham, 2014; Springenberg et al., 2014) have shown impressive results by performing heavier data augmentation on higher resolution versions on CIFAR-10. We plan to extend this analysis to higher resolution data sets such as ImageNet and one could expect even more benefits from data augmentation compared to explicit regularization techniques.

## ACKNOWLEDGMENTS

This project has received funding from the European Union's Horizon 2020 research and innovation programme under the Marie Sklodowska-Curie grant agreement No 641805.

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

# A    DETAILED AND EXTENDED EXPERIMENTAL RESULTS

This appendix details the results of the main experiments shown in Figure 2 and provides the results of many other experiments. For example, the results of the models trained with dropout, but without weight decay and the results of training with 80 % and 1 % of the data are not shown in Figure 2 in order not to clutter the visualization.

Additionally, for most experiments we train a version of the network without batch normalization. These results are provided within brackets in the tables. Note that the original All-CNN results published by Springenberg et al. (2014) did not include batch normalization. In the case of WRN, we remove all batch normalization layers except the top-most one, before the spatial average pooling, since otherwise many models would not converge.

Table 2: Test accuracy of the networks All-CNN and WRN on CIFAR-10 and CIFAR-100, comparing the performance with and without explicit regularizers and the different augmentation schemes. Results within brackets show the performance of the models without batch normalization

| Network | WD | Dropout | Aug. scheme | Test CIFAR-10 | Test CIFAR-100 |
|---------|-----|---------|-------------|---------------|----------------|
|         | yes | yes | no | 90.04 (88.35) | 66.50 (60.54) |
|         | yes | yes | light | 93.26 (91.97) | 70.85 (65.57) |
|         | yes | yes | heavier | 93.08 (92.44) | 70.59 (68.62) |
|         | no | yes | no | 77.99 (87.59) | 52.39 (60.96) |
| All-CNN | no | yes | light | 77.20 (92.01) | 69.71 (68.01) |
|         | no | yes | heavier | 88.29 (92.18) | 70.56 (68.40) |
|         | no | no | no | 84.53 (71.98) | 57.99 (39.03) |
|         | no | no | light | 93.26 (90.10) | 69.26 (63.00) |
|         | no | no | heavier | 93.55 (91.48) | 71.25 (71.46) |
|         | yes | yes | no | 91.44 (89.30) | 71.67 (67.42) |
|         | yes | yes | light | 95.01 (93.48) | 77.58 (74.23) |
|         | yes | yes | heavier | 95.60 (94.38) | 76.96 (74.79) |
|         | no | yes | no | 91.47 (89.38) | 71.31 (66.85) |
| WRN     | no | yes | light | 94.76 (93.52) | 77.42 (74.62) |
|         | no | yes | heavier | 95.58 (94.52) | 77.47 (73.96) |
|         | no | no | no | 89.56 (85.45) | 68.16 (59.90) |
|         | no | no | light | 94.71 (93.69) | 77.08 (75.27) |
|         | no | no | heavier | 95.47 (94.95) | 77.30 (75.69) |

An important observation from Table 2 is that the interaction of weight decay and dropout is not always consistent, since in some cases better results are obtained with both explicit regularizers active and in other cases, only dropout achieves better generalization. However, the effect of data augmentation seems to be clearer: just some light augmentation achieves much better results than training only with the original data set and performing heavier augmentation almost always further improves the test accuracy, without the need for explicit regularization.

Not surprisingly, batch normalization also contributes to improve the generalization of All-CNN and it seems to combine well with data augmentation. On the contrary, when combined with explicit regularization the results are interestingly not consistent in the case of All-CNN: it seems to improve the generalization of the model trained with both weight decay and dropout, but it drastically reduces the performance with only dropout, in the case of CIFAR-10 and CIFAR-100 without augmentation. A probable explanation is, again, that the regularization hyperparameteres would need to be readjusted with a change of the architecture.

Furthermore, it seems that the gap between the performance of the models trained with and without batch normalization is smaller when they are trained without explicit regularization and when they include heavier data augmentation. This can be observed in both Table 2 and Table 3, which contains the results of the models trained with fewer examples. It is important to note as well the benefits of batch normalization for obtaining better results when training with fewer examples. However, it is surprising that there is only a small drop in the performance of WRN—95.47 % to 94.95 % without regularization— from removing the batch normalization layers of the residual blocks, given

that they were identified as key components for training deep residual networks (He et al., 2016; Zagoruyko & Komodakis, 2016).

Table 3: Test accuracy when training with only 80 %, 50 %, 10 % and 1 % of the available training examples. Results within brackets show the performance of the models without batch normalization

| Network | Pct. Data | Explicit Reg. | Aug. scheme | Test CIFAR-10 | Test CIFAR-100 |
|---------|-----------|---------------|-------------|---------------|----------------|
| All-CNN | 80 % | yes | no | 89.41 (86.61) | 63.93 (52.51) |
|  |  | yes | light | 92.20 (91.25) | 67.63 (63.24) |
|  |  | yes | heavier | 92.83 (91.42) | 68.01 (65.89) |
|  |  | no | no | 83.04 (75.00) | 55.78 (35.95) |
|  |  | no | light | 92.25 (88.75) | 69.05 (56.81) |
|  |  | no | heavier | 92.80 (90.55) | 69.40 (63.57) |
|  | 50 % | yes | no | 85.88 (82.33) | 58.24 (44.94) |
|  |  | yes | light | 90.30 (87.37) | 61.03 (54.68) |
|  |  | yes | heavier | 90.09 (88.94) | 63.25 (57.91) |
|  |  | no | no | 78.61 (69.46) | 48.62 (31.81) |
|  |  | no | light | 90.21 (84.38) | 62.83 (47.84) |
|  |  | no | heavier | 90.76 (87.44) | 64.41 (55.27) |
|  | 10 % | yes | no | 67.19 (61.61) | 33.77 (19.79) |
|  |  | yes | light | 76.03 (69.18) | 38.51 (22.79) |
|  |  | yes | heavier | 78.69 (64.14) | 38.34 (26.29) |
|  |  | no | no | 60.97 (41.07) | 26.05 (17.55) |
|  |  | no | light | 78.29 (67.65) | 37.84 (24.34) |
|  |  | no | heavier | 79.87 (70.64) | 39.85 (26.31) |
|  | 1 % | yes | no | 27.53 (29.90) | 9.16 (3.60) |
|  |  | yes | light | 37.18 (26.85) | 9.64 (3.65) |
|  |  | yes | heavier | 42.73 (26.87) | 9.14 (2.52) |
|  |  | no | no | 38.89 (35.68) | 9.50 (5.51) |
|  |  | no | light | 44.35 (29.29) | 9.87 (5.36) |
|  |  | no | heavier | 47.60 (33.72) | 11.45 (3.57) |
| WRN | 80 % | yes | no | 90.27 | 70.41 |
|  |  | yes | light | 94.07 | 75.66 |
|  |  | yes | heavier | 94.57 | 75.51 |
|  |  | no | no | 88.98 | 66.10 |
|  |  | no | light | 93.97 | 75.07 |
|  |  | no | heavier | 94.84 | 75.38 |
|  | 50 % | yes | no | 86.96 | 63.60 |
|  |  | yes | light | 92.65 | 70.83 |
|  |  | yes | heavier | 92.86 | 70.33 |
|  |  | no | no | 85.56 | 60.64 |
|  |  | no | light | 91.87 | 69.97 |
|  |  | no | heavier | 92.77 | 70.72 |
|  | 10 % | yes | no | 70.73 | 34.11 |
|  |  | yes | light | 76.00 | 36.65 |
|  |  | yes | heavier | 78.10 | 38.93 |
|  |  | no | no | 60.39 | 23.65 |
|  |  | no | light | 79.19 | 39.24 |
|  |  | no | heavier | 80.29 | 41.44 |
|  | 1 % | yes | no | 33.45 | 7.47 |
|  |  | yes | light | 34.13 | 7.50 |
|  |  | yes | heavier | 41.02 | 8.37 |
|  |  | no | no | 38.63 | 9.47 |
|  |  | no | light | 43.84 | 9.91 |
|  |  | no | heavier | 47.14 | 11.03 |

The results in Table 3 clearly support the conclusion presented in Section 2.3: data augmentation alone resists better the lack of training data compared to explicit regularizers. Already with 80% and

Table 4: Test accuracy of the shallower and deeper versions of the All-CNN network on CIFAR-10 and CIFAR-100. Results in parentheses show the difference with respect to the original model.

| Network | Explicit Reg. | Aug. scheme | Test CIFAR-10 | Test CIFAR-100 |
|---|---|---|---|---|
| | yes | no | 76.45 (-13.59) | 51.31 (-9.23) |
| | yes | light | 82.02 (-11.24) | 56.81 (-8.76) |
| All-CNN shallower | yes | heavier | 86.66 (-6.42) | 58.64 (-9.98) |
| | no | no | 85.22 (+0.69) | 58.95 (+0.96) |
| | no | light | 90.02 (-3.24) | 65.51 (-3.75) |
| | no | heavier | 90.34 (-3.21) | 65.87 (-5.38) |
| | yes | no | 86.26 (-3.78) | 49.06 (-11.48) |
| | yes | light | 85.04 (-8.22) | 52.03 (-13.54) |
| All-CNN deeper | yes | heavier | 88.46 (-4.62) | 51.78 (-16.84) |
| | no | no | 83.30 (-1.23) | 54.22 (-3.77) |
| | no | light | 93.46 (+0.20) | 72.16 (+2.90) |
| | no | heavier | 94.19 (+0.64) | 73.30 (+2.35) |

50% of the data better results are obtained in some cases, but the differences become much bigger when training with only 10% and 1% of the available data. It seems that explicit regularization prevents the model from both fitting the data and generalizing well, whereas data augmentation provides useful transformed examples. Interestingly, with only 1% of the data, even without data augmentation the models without explicit regularization perform better.

The same effect can be observed in Table 4, where both the shallower and deeper versions of All-CNN perform much worse when trained with explicit regularization, even when trained without data augmentation. This is another piece of evidence that explicit regularization needs to be used very carefully, it requires a proper tuning of the hyperparameters and is not always benefitial.

## B    NORM OF THE WEIGHT MATRIX

Table 5: Frobenius norm of the weight matrices learned by the networks All-CNN and WRN on CIFAR-10 and CIFAR-100, trained with and without explicit regularizers and the different augmentation schemes. Norms within brackets correspond to the models without batch normalization

| Network | WD | Dropout | Aug. scheme | Norm CIFAR-10 | Norm CIFAR-100 |
|---|---|---|---|---|---|
| | yes | yes | no | 48.7 (64.9) | 76.5 (97.9) |
| | yes | yes | light | 52.7 (63.2) | 77.6 (86.8) |
| | yes | yes | heavier | 57.6 (62.8) | 78.1 (83.1) |
| | no | yes | no | 52.4 (70.5) | 79.7 (103.3) |
| All-CNN | no | yes | light | 57.0 (67.9) | 83.6 (93.0) |
| | no | yes | heavier | 62.8 (67.5) | 84.0 (88.0) |
| | no | no | no | 37.3 (63.7) | 47.6 (102.7) |
| | no | no | light | 47.0 (69.5) | 80.0 (108.9) |
| | no | no | heavier | 62.0 (71.7) | 91.7 (91.7) |
| | yes | yes | no | 101.4 (122.6) | 134.8 (126.5) |
| | yes | yes | light | 106.1 (123.9) | 140.8 (129.3) |
| | yes | yes | heavier | 119.3 (125.3) | 164.2 (132.5) |
| | no | yes | no | 153.3 (122.5) | 185.1 (126.5) |
| WRN | no | yes | light | 160.6 (123.9) | 199.0 (129.4) |
| | no | yes | heavier | 175.1 (125.2) | 225.4 (132.5) |
| | no | no | no | 139.0 (120.4) | 157.9 (122.0) |
| | no | no | light | 153.6 (123.2) | 187.0 (127.2) |
| | no | no | heavier | 170.4 (125.4) | 217.6 (132.9) |

One of the simplest way of getting a rough idea of the complexity of the learned models is computing the norm of the weight matrix. Table 5 shows the Frobenius norm of the weight matrices of

Table 6: Frobenius norm of the weight matrices learned by the shallower and deeper versions of the All-CNN network on CIFAR-10 and CIFAR-100.

| Network | Explicit Reg. | Aug. scheme | Norm CIFAR-10 | Norm CIFAR-100 |
|---|---|---|---|---|
| All-CNN shallower | yes | no | 47.9 | 68.9 |
| | yes | light | 49.7 | 67.1 |
| | yes | heavier | 51.9 | 66.2 |
| | no | no | 34.8 | 64.7 |
| | no | light | 45.6 | 68.8 |
| | no | heavier | 53.1 | 68.3 |
| All-CNN deeper | yes | no | 62.3 | 92.1 |
| | yes | light | 66.5 | 95.7 |
| | yes | heavier | 71.5 | 96.9 |
| | no | no | 45.4 | 53.4 |
| | no | light | 57.3 | 77.3 |
| | no | heavier | 70.7 | 97.5 |

the models trained with different levels of explicit regularization and data augmentation. The clearest conclusion is that heavier data augmentation seems to yield solutions with larger norm. This is always true except in some All-CNN models trained without batch normalization. Another observation is that, as expected, weight decay constrains the norm of the learned function. Besides, the models trained without batch normalization exhibit smaller differences between different levels of regularization and augmentation and, in the case of All-CNN, less consistency.

One of the relevant results presented in this paper is the poor performance of the regularized models on the shallower and deeper versions of All-CNN, compared to the models without explicit regularization (see Table 4). One hypothesis is that the *amount* of regularization is not properly adjusted through the hyperparameters. This could be reflected in the norm of the learned weights, shown in Table 6. However, the norm alone does not seem to fully explain the large performance differences between the different models. Finding the exact reasons why the regularized models not able to generalize well might require a much thourough analysis and we leave it as future work.

