# OpenReview forum: "Data augmentation instead of explicit regularization"
_ICLR.cc/2018/Conference — Reject_

### Official Review · AnonReviewer1 · 2017-11-17
**Systematic study of data augmentation in image classification problems**

**Rating:** 5
**Confidence:** 4

**Review:**

This paper provides a systematic study of data augmentation in image classification problems with deep neural networks and argues that data augmentation could replace some common explicit regularizers like the weight decay and dropout. The data augmentation techniques are also shown to be insensitive to hyper parameters, so easier to use than explicit regularizers when changing architectures.

It is good to have a systematic study of data augmentations, however, the materials in this paper in the current state might not be a strong ICLR publication. The paper could potentially be made more interesting or solid if some of the followings could be investigated:

- considering a wider range of different problems apart from image classification, and investigate the effectiveness of domain specific data augmentation and general data augmentation
- systematically study each of the data augmentation techniques separately to see which is more important (as oppose to only having 'light' and 'heavy' scheme); potentially also study other less-traditional augmentation schemes such as adversarial examples, etc.
- propose novel data augmentation schemes
- more analysis of the interplay with Batch Normalization, why the results for BN vs no-BN is not presented for WRN?
- carefully designed synthetic (or real) data / task to verify the statements. For example, the explicit regularizers are thought to unnecessarily constraint the model too much. Can you measure the norm (or other complexity measures) of the models learned with explicit regularizers vs models learned with data augmentation?

---

> ### Author Response · Authors · 2018-01-03
> **Incorporation and discussion of the improvements suggested by the reviewer**
>
> We would first like to thank the reviewer for the feedback. In particular, we are grateful for the suggestions to make a stronger paper, which we have gladly received and added to our work. Besides, we highly appreciate the accurate summary of the paper, where the key points of our paper are correctly identified. That reflects a careful read of the paper.
>
> The reviewer suggests some extensions for the work that would make the paper “more interesting or solid”. We comment on them next:
>
> - Complexity measures: we fully agree with the reviewer and we have accordingly added an appendix section (B) with an analysis of the Frobenius norm of the learnt weights by the models with different level of regularization and data augmentation. The main conclusions are: a) more augmentation yields solutions of larger norm b) more regularization yields solutions of smaller norm. This is in line with the hypotheses presented in the paper.
>
> - More analysis of the interplay with Batch Normalization: the full report of the results from experiments with and without batch normalization, together with the corresponding analysis, is now provided in the Appendix A. We also include new experiments that we hope will complement the already included ones. They had not been included in the original version due to space limitations, but we have rearranged the paper and included some appendices with the hope of making the paper more solid, according to the reviewer suggestion.
>
> - Novel data augmentation schemes: the proposal of new augmentation is schemes is not the aim of this paper, but rather analyzing the interplay of (general) data augmentation and regularization. We believe that the selected schemes are representative of common practices in the literature and of different levels of augmentation. We also think that there is no need for proposing new schemes, as there exist many in the literature. Several of them are referenced in our paper:  (Graham, 2014), (DeVries & Taylor, 2017a),  (DeVries & Taylor, 2017b), etc. Besides, we think that in the near future it will be possible to automatically learn the augmentation (Lemley et al., 2017; Ratner et al., 2017)
>
> - Systematic study of the augmentation schemes: in line with the previous suggestion, we believe that such an analysis is out of the scope. We aimed at analyzing data augmentation as a general concept/technique, rather than studying the individual types of augmentation.
>
> - Consider a wider range of different problems apart from image classification: although it would be very interesting to see the same analysis on other type of problems, the amount of work would increase very significantly and cannot be in the scope of a single paper, in our opinion. Furthermore and very importantly, we cannot assume that the same analysis and results would be easily transferred to other domains, because we cannot disregard the role of implicit regularization provided by convolutional layers, for instance, which are essential for the particular task of object recognition.

---

### Official Review · AnonReviewer2 · 2017-11-28
**Detailed study of data augmentation is a great idea, but more variety is needed to draw meaningful conclusions**

**Rating:** 5
**Confidence:** 4

**Review:**

The paper proposes data augmentation as an alternative to commonly used regularisation techniques like weight decay and dropout, and shows for a few reference models / tasks that the same generalization performance can be achieved using only data augmentation.

I think it's a great idea to investigate the effects of data augmentation more thoroughly. While it is a technique that is often used in literature, there hasn't really been any work that provides rigorous comparisons with alternative approaches and insights into its inner workings. Unfortunately I feel that this paper falls short of achieving this.

Experiments are conducted on two fairly similar tasks (image classification on CIFAR-10 and CIFAR-100), with two different network architectures. This is a bit meager to be able to draw general conclusions about the properties of data augmentation. Given that this work tries to provide insight into an existing common practice, I think it is fair to expect a much stronger experimental section. In section 2.1.1 it is stated that this was a conscious choice because simplicity would lead to clearer conclusions, but I think the conclusions would be much more valuable if variety was the objective instead of simplicity, and if larger-scale tasks were also considered.

Another concern is that the narrative of the paper pits augmentation against all other regularisation techniques, whereas more typically these will be used in conjunction. It is however very interesting that some of the results show that augmentation alone can sometimes be enough.

I think extending the analysis to larger datasets such as ImageNet, as is suggested at the end of section 3, and probably also to different problems than image classification, is going to be essential to ensure that the conclusions drawn hold weight.



Comments:

- The distinction between "explicit" and "implicit" regularisation is never clearly enunciated. A bunch of examples are given for both, but I found it tricky to understand the difference from those. Initially I thought it reflected the intention behind the use of a given technique; i.e. weight decay is explicit because clearly regularisation is its primary purpose -- whereas batch normalisation is implicit because its regularisation properties are actually a side effect. However, the paper then goes on to treat data augmentation as distinct from other explicit regularisation techniques, so I guess this is not the intended meaning. Please clarify this, as the terms crop up quite often throughout the paper. I suspect that the distinction is somewhat arbitrary and not that meaningful.

- In the abstract, it is already implied that data augmentation is superior to certain other regularisation techniques because it doesn't actually reduce the capacity of the model. But this ignores the fact that some of the model's excess capacity will be used to model out-of-distribution data (w.r.t. the original training distribution) instead. Data augmentation always modifies the distribution of the training data. I don't think it makes sense to imply that this is always preferable over reducing model capacity explicitly. This claim is referred to a few times throughout the work.

- It could be more clearly stated that the reason for the regularising effect of batch normalisation is the noise in the batch estimates for mean and variance.

- Some parts of the introduction could be removed because they are obvious, at least to an ICLR audience (like "the model would not be regularised if alpha (the regularisation parameter) equals 0").

- The experiments with smaller dataset sizes would be more interesting if smaller percentages were used. 50% / 80% / 100% are all on the same order of magnitude and this setting is not very realistic. In practice, when a dataset is "too small" to be able to train a network that solves a problem reliably, it will generally be one or more orders of magnitude too small, not 2x too small.

- The choices of hyperparameters for "light" and "heavy" motivation seem somewhat arbitrary and are not well motivated. Some parameters which are sampled uniformly at random should be probably be sampled log-uniformly instead, because they represent scale factors. It should also be noted that much more extreme augmentation strategies have been used for this particular task in literature, in combination with padding (for example by Graham). It would be interesting to include this setting in the experiments as well.

- On page 7 it is stated that "when combined with explicit regularization, the results are much worse than without it", but these results are omitted from the table. This is unfortunate because it is a very interesting observation, that runs counter to the common practice of combining all these regularisation techniques together (e.g. L2 + dropout + data augmentation is a common combination). Delving deeper into this could make the paper a lot stronger.

- It is not entirely true that augmentation parameters depend only on the training data and not the architecture (last paragraph of section 2.4). Clearly more elaborate architectures benefit more from data augmentation, and might need heavier augmentation to perform optimally because they are more prone to overfitting (this is in fact stated earlier on in the paper as well). It is of course true that these hyperparameters tend to be much more robust to architecture changes than those of other regularisation techniques such as dropout and weight decay. This increased robustness is definitely useful and I think this is also adequately demonstrated in the experiments.

- Phrases like "implicit regularization operates more effectively at capturing reality" are too vague to be meaningful.

- Note that weight decay has also been found to have side effects related to optimization (e.g. in "Imagenet classification with deep convolutional neural networks", Krizhevsky et al.)

REVISION: I applaud the effort the authors have put in to address many of my and the other reviewers' comments. I think they have done so adequately for the most part, so I've decided to raise the rating from 3 to 5, for what it's worth.

The reason I have decided not to raise it beyond that, is that I still feel that for a paper like this, which studies an existing technique in detail, the experimental side needs to be significantly stronger. While ImageNet experiments may be a lot of work, some other (smaller) additional datasets would also have provided more interesting evidence. CIFAR-10 and CIFAR-100 are so similar that they may as well be considered variants of the same dataset, at least in the setting where they are used here.

I do really appreciate the variety in the experiments in terms of network architectures, regularisation techniques, etc. but I think for real-world relevance, variety in problem settings (i.e. datasets) is simply much more important. I think it would be fine if additional experiments on other datasets were not varied along all these other axes, to cut down on the amount of work this would involve. But not including them at all unfortunately makes the results much less impactful.

---

> ### Author Response · Authors · 2018-01-03
> **Suggestions by the reviewer incorporated to the new version.**
>
> We thank the reviewer for the very useful feedback, in particular for the interesting suggestions we have adopted to hopefully make our paper stronger. We especially thanks the words appreciating the usefulness of our study.
>
> In the following we will answer the particular comments of the reviewer, given as a list:
>
> - [Distinction between explicit and implicit regularization]: this is a very good point and we thank the reviewer for the suggestion of providing a definition. We have updated the paper and we now provide a definition and distinction of explicit and implicit regularization in the introduction on page 1.
>
> - [“Data augmentation always modifies the distribution of the training data”]: As the reviewer indicates, some of the model’s capacity is used to model out-of-distribution data. However, these out-of-distribution data are generated in a plausible way, i.e. the augmentation schemes are designed so the generated examples reflect plausible variations of the original distribution present in the real world, as opposed to, for instance, dropout, which randomly turns off neurons during training.
>
> - [State the reason for the regularising effect of batch normalisation]: we appreciate the suggestion. We have added a sentence about this on the second-to-last paragraph on page 1.
>
> - [Obvious parts for an ICLR audience in the introduction]: We agree and have actually re-written the first part of the introduction. Instead of explaining the concept of regularization and highlighting its importance in machine learning, we now directly and briefly define regularization, then highlight its particular role on deep learning and finally provide our definition of the concepts of explicit and implicit regularization.
>
> - [Experiments with even smaller data set sizes]: we highly appreciate this suggestion. In the original version we only provided results with 80 and 50 % of the data. In the new version of the paper we now provide results with 10 and 1 %, additionally. The results with even smaller data set sizes actually strengthen the benefits of data augmentation with respect to explicit regularisation. We invite the readers to check the new results on the graphical visualization on page 6, the analysis on Section 2.3 and the full report of results in the Appendix A.
>
> - [“The choices of hyperparameters for "light" and "heavy" motivation seem somewhat arbitrary and are not well motivated.”]: As stated by the reviewer, the choices are indeed very arbitrary. However, we see this as a good thing, in contrast to regularization hyperparameters, which cannot be arbitrary at all, but very well chosen. One of the advantages of data augmentation with respect to regularization is that it does not require careful selection and validation of the hyperparameters. The schemes suggested by the reviewer are actually referenced in the paper. However, our goal is not performing a review of data augmentation strategies used in the literature, but showing the advantage of data augmentation as a general concept/technique over regularisation. The suggestion of sampling log-uniformly would have perhaps made more sense, but we consider this is not crucial, and the fact that we have (deliberately) designed a suboptimal data augmentation strategy better supports our hypothesis, and one could expect even more benefits from data augmentation compared to explicit regularization with more carefully designed augmentation schemes.
>
> - [“On page 7 it is stated that "when combined with explicit regularization, the results are much worse than without it", but these results are omitted from the table. This is unfortunate because it is a very interesting observation [...] Delving deeper into this could make the paper a lot stronger.”]: It should be noted that this sentence refers to the results with *batch normalization*, not to explicit regularization + data augmentation, which were indeed already provided and accordingly analyzed. Nonetheless, regarding the results analyzing the use of batch normalization, please note that we have extended the report of results in the Appendix A.
>
> - [“It is not entirely true that augmentation parameters depend only on the training data and not the architecture”]: This is a good point and we have accordingly reduced the scope of that claim (last paragraph of Section 2.4).
>
> - [“Phrases like "implicit regularization operates more effectively at capturing reality" are too vague to be meaningful”]: We have accordingly updated the fourth paragraph of Section 3.
>
> - [Optimization side-effects of weight decay]: This has been in fact observed by many researchers, included ourselves, although not always reported. We humbly think that it even supports our proposal of not overusing weight decay, since modern deep architectures, trained with batch normalization, SGD and appropriate typically do not present any optimization problems.

---

> ### Author Response · Authors · 2018-01-03
> **Discussion of main concerns of the reviewer**
>
> The main concern of the reviewer is the lack of experiments on a wider variety of data sets, such as ImageNet. We would also, of course, like to be able of doing the same analysis on ImageNet and other data sets. However, we reckon that one of the strengths of our paper is that we analyze the differences between data augmentation and explicit regularization under a wide variety of factors: 2 different network architectures, 3 levels of data augmentation, 3 types of regularization, 3 levels of network depth, 2 data sets, 4 training data sizes, with and without batch normalization, etc. Applying the same amount of experiments on ImageNet is unfortunately unfeasible. As a matter of fact, the vast majority of papers on object recognition doing equivalent analysis (including most ICLR papers) limit their analysis to CIFAR, MNIST and SVHN. Since we have found a high degree of consistency in our experiments, we humbly believe that one can expect similar results on ImageNet (and probably even better benefits from data augmentation vs. explicit regularization due to the higher resolution). Nonetheless, we plan to extend the analysis to ImageNet, to a certain extent, in future works. Finally, please note that we acknowledge and comment on this limitation in the discussion section of the paper.
>
> Another concern of the reviewer is that “the narrative of the paper pits augmentation against all other regularisation techniques, whereas more typically these will be used in conjunction.” In this regard, we deliberately propose not using explicit regularization techniques in the cases where sufficient data augmentation can be applied. In view of our results (please, see the updated version of the paper, with an improved visualization of the results on page 6 and the extended report in the appendix A), in the vast majority of cases data augmentation alone can achieve at least the same performance or even better than augmentation+regularization, as actually noted by the reviewer. Thus, why using them in conjunction. Dropping explicit regularization techniques such as dropout and weight decay has a number of advantages: faster training, better interpretable models, fewer hyperparameters, etc.

---

### Official Review · AnonReviewer3 · 2017-11-28
**Interesting empirical study but unclear outcome and results do not seem to support the conclusions**

**Rating:** 5
**Confidence:** 4

**Review:**

This paper presents an empirical study of whether data augmentation can be a substitute for explicit regularization of weight decay and dropout.  It is a well written and well organized paper.  However, overall I do not find the authors’ premises and conclusions to be well supported by the results and would suggest further investigations.  In particular:

a) Data augmentation is a very domain specific process and limits of augmentation are often not clear.  For example, in financial data or medical imaging data it is often not clear how data augmentation should be carried out and how much is too much.  On the other hand model regularization is domain agnostic (has to be tuned for each task, but the methodology is consistent and well known).  Thus advocating that data augmentation can universally replace explicit regularization does not seem correct.

b) I find the results to be somewhat inconsistent.  For example, on CIFAR-10, for 100% data regularization+augmentation is better than augmentation alone for both models, whereas for 80% data augmentation alone seems to be better.  Similarly on CIFAR-100 the WRN model shows mixed trends, and this model is significantly better than the All-CNN model in performance.  These results also seem inconsistent with authors statement “…and conclude that data augmentation alone - without any other explicit regularization techniques - can achieve the same performance to higher as regularized models…”

---

> ### Author Response · Authors · 2017-12-30
> **The results do support the conclusions but were not presented clearly enough. We add new visual presentation of results.**
>
> We thank the reviewer for the insightful comments and for raising some concerns that we have positively received and used to, in our opinion, improve the paper.
>
> The main concern is that “the results do not seem to support the conclusions”. We humbly believe that our results do support the conclusions, but we agree that they might not “seem” so because we failed at presenting the results in a more clear way in the original version of the paper.
>
> We invite you to check the new version of the paper, where the main results are now presented graphically in page 6, enabling easier comparisons of the models with and without explicit regularization and the different augmentation schemes. We have moved the tables with the full details of all experiments to the appendix. We hope it is now easier to see how the results support the conclusions: data augmentation alone provides at least the same performance than data augmentation + explicit regularization in the vast majority of the experiments.
>
> Regarding your particular comments:
>
> a) Although data augmentation is indeed domain specific and requires certain expert knowledge, we believe that 1) it should not be a reason to disregard it. Instead, expert knowledge should be exploited. Deep learning is full of very successful domain-specific techniques: convolutional layers, LSTM, etc. 2) one data augmentation scheme can be designed for a broad set of tasks (for example, object recognition, segmentation, localisation) and a broad family of data (natural images) and it will naturally apply to many types of architectures, hyperparameters, amount of training data, tasks, data sets, etc, as supported in our experiments.
>
> In no place do we claim that data augmentation can “universally” replace regularization. Our results apply only, of course, to the cases where enough data augmentation can be applied. There exists a significantly large corpus of works on financial data and medical imaging, for example, where data augmentation is also applied and we believe that future research will bring more advanced techniques in this direction.
>
> Explicit regularization is well-known only to some extent though and the disadvantages are multiple: notably, the need for specific hyperparameter tuning for every task, every architecture, every amount of training data, etc. A piece of evidence is that new regularization techniques are continuously proposed in the literature.
>
> Thus, we propose avoiding explicit regularization when it is unnecessary (when it can be substituted by data augmentation).
>
> b) For CIFAR10, All-CNN and 100 % of the data: only data augmentation (93.55) outperforms data augmentation + regularization (93.08). In the case of WRN, both cases are equivalent (95.47 vs 95.60). Similar results can be found in the rest of the experiments, both on CIFAR10 and CIFAR100 and the difference in favour of only data augmentation increases as fewer examples are used for training. We gladly invite the readers to have a look at Figure 2 and the tables of the appendix to check these results.
>
> As a final remark, we find important to note that in order to enable fair comparisons, we haven’t optimized the learning rate (nor other training hyperparameters) of our proposed way of training (only data augmentation, no explicit regularization), in contrast to the highly optimized set of hyperparameters provided in a paper that achieved state-of-the-art results and was published in ICLR 2014 (All-CNN). We used their original training hyperparameters, which were optimized for a model that includes weight decay, dropout, and no batch norm, which is highly suboptimal. As a matter of fact, we have observed that if the learning rate is increased, data augmentation alone achieves higher results than the ones we provide in the paper.

---

### Author Response · Authors · 2018-01-03
**Summary of changes**

Inspired by the reviewers' feedback, we have improved our paper with a number of changes. Here is a summary of the main improvements:

- Graphical visualization of the main results - page 6. In the original version, all the results were presented only numerically in tables. The newer version presents the most important results in a graphical way, using color bars, which hopefully enables an easier comparisons of the different experiments. [From reviewer 3]

- Detailed report of the results in the appendix - Appendix A, pages 12-14. Since the main results are now presented in a graphical way in the main body of the paper, we have moved the tables with the detailed report of the results to an appendix. [From reviewers 1, 2 and 3]

- New experiments: results of training with and without batch normalization - Appendix A, pages 12-14. In the original version, only a subset of the experiments comparing the models trained with and without batch normalization were reported. The newer version reports both versions (with and without) for most of the models under test. [From reviewer 1]

- New experiments: training with reduced data sets, 10 and 1 % of the data - Appendix A, page 13. Whereas the original version reports only the results of training with 80 and 50 % of the data, the newer version reports a wider set of experiments: 80, 50, 10 and 1 % of the training data. [From reviewer 2]

- New analysis: norm of the weight matrix - Appendix B, pages 14-15. The analysis of the norm of the weight matrix provides a way to compare the complexity of the function learned by models trained with different levels of regularization and data augmentation. [From reviewer 1]

- Definitions of explicit and implicit regularization - Introduction, page 1. In order to reduce the ambiguity and facilitate the understanding of the paper, we provide our definitions of the concepts of explicit and implicit regularization, which are unfortunately not clearly defined in the literature. [From reviewer 2]

We would like to sincerely thank again the reviewers for their useful feedback.

---

### Decision · Program_Chairs · 2018-01-29
**ICLR 2018 Conference Acceptance Decision**

**Decision:**

Reject

**Comment:**

The reviewers agree that the authors have made an interesting contribution studying the effect of data augmentation, but they also agree that the claims made by the paper require a broader empirical study beyond the limited number of tasks surveyed in the current revision.  I urge the authors to follow this advice and see what they find.